# A Multi-Feature Search Window Method for Road Boundary Detection Based on LIDAR Data

**DOI:** 10.3390/s19071551

**Published:** 2019-03-30

**Authors:** Kai Li, Jinju Shao, Dong Guo

**Affiliations:** School of Transportation and Vehicle Engineering, Shandong University of Technology, Zibo 255000, Shandong, China; likai_academic@163.com (K.L.); shaojinju@sdut.edu.cn (J.S.)

**Keywords:** structured road, LIDAR point cloud, multi-feature extraction, boundary detection

## Abstract

In order to improve the accuracy of structured road boundary detection and solve the problem of the poor robustness of single feature boundary extraction, this paper proposes a multi-feature road boundary detection algorithm based on HDL-32E LIDAR. According to the road environment and sensor information, the former scenic cloud data is extracted, and the primary and secondary search windows are set according to the road geometric features and the point cloud spatial distribution features. In the search process, we propose the concept of the largest and smallest cluster points set and a two-way search method. Finally, the quadratic curve model is used to fit the road boundary. In the actual road test in the campus road, the accuracy of the linear boundary detection is 97.54%, the accuracy of the curve boundary detection is 92.56%, and the average detection period is 41.8 ms. In addition, the algorithm is still robust in a typical complex road environment.

## 1. Introduction

At present, the automatic driving is mainly realized by Global Position System (GPS) positioning tracing and on-board sensor sensing. The GPS is an indispensable technology for current driving positioning and plays a very important role in autonomous vehicles positioning. In the research on autonomous vehicles, Wu et al. [1] design a real-time autonomous navigation system by integrating a real-time kinematics differential global position system (RTK-DGPS). The main destination of the AUTOPIA program [2,3] in Europe is to develop autonomous vehicles by using GPS sensors for navigation and fuzzy logic for lateral control. Lundgren et al. [4] presented a localization algorithm based on a map and a set of off-the-shelf sensors, with the purpose of evaluating a low-cost solution with respect to localization performance. In addition, GPS data is also essential to improve map information. For example, in order to enrich the information of OpenStreetMap, Mobasheri et al. [5] constructed sidewalk geometries by mining data from GPS data of people in wheelchairs. However, GPS positioning is usually affected by factors such as weather and building occlusion, causing signal drift or loss. At this time, it is necessary to divide the road area and the non-road area by road boundary detection to determine the safe driving area of the intelligent vehicle [6], and at the same time reduce the sensor search range to improve the sensing accuracy and real-time performance [7,8]. As the urbanization process continues to accelerate and the degree of structuralism of roads is becoming more and more perfect, the boundary detection of structured roads has always been an important part of the perception of automatic driving.

Most of the structured road detection methods are based on cameras and two-dimensional LIDAR to obtain environmental information [9,10,11,12], and the driving area is segmented by color, texture and other features to achieve the extraction of road boundaries. Rami et al. [13] proposed a novel detection algorithm for vision systems based on combined fuzzy image processing and bacterial algorithm, which has been modified and applied to produce a time-based trajectory for the optimal path. Seibert et al. [14] described the extension of an existing commercial lane detection system for marked roads by the detection of soft shoulders, curbs and guardrails based on a front directed gray scale camera. Zhang et al. [15] used a tracking camera to propose a local adaptive threshold image binary segmentation algorithm for path recognition, and then accurately extracted the strip centerline for path tracking to accurately complete the unmanned vehicle path tracking. Oniga and Nedevschi [16] proposed a real-time algorithm for curb detection based on dense stereovision, and transformed the 3D points from stereovision into a Digital Elevation Map. The method based on image segmentation can obtain rich road information, but the method is greatly affected by environmental factors such as illumination, rain and fog, and the real-time performance of the algorithm is difficult to meet due to the large amount of information processing. Two-dimensional LIDAR mainly extracts road boundary candidate points by a certain height of the road shape and then achieves road boundary fitting [17,18,19]. Nan et al. [20] developed a road boundary detection algorithm to extract road features as line segments in polar coordinates relative to the 2D Lidar sensor. In order to create a map based on Lidar in GPS-denied environments, Kang et al. [21] proposed a high precision 2D laser point-clouds map creating method based on loop closure detection. Liu et al. [22] used 2D sequential laser range finder data and vehicle state information to build a local Digital Elevation Map (DEM), and achieved curb detection with 1D Gaussian process regression. However, with the increase of the vehicle speed, the rate of missed detection also increases, and it is greatly affected by roadside vegetation. 

With the development of 3D point cloud data acquisition technology, road boundary detection based on 3D LIDAR has gradually become one of the research hotspots at home and abroad [23,24,25,26]. In order to improve the robustness of detection, Liu et al. [27] established a virtual LIDAR model based on HDL-64E LIDAR. Chen et al. [28] extracted the boundary seed points from the 3D point cloud by distance constraint, and increased the number of seed points by region growth, and finally performed a quadratic curve fitting to obtain the complete road boundary shape. Wang et al. [29] extracted the road boundary candidate points from each LIDAR scan result by Hough transform, and fitted the road boundary by an iterative Gaussian process regression. Zhang et al. [30] segmented the travelable area according to the structural road height difference jump characteristics, and divided the left and right non-road areas through the obstacle grid map. In Su et al. [31], in order to avoid the occurrence of pseudo-boundary interference in point cloud data processing, the adaptive direction boundary search algorithm is used to extract the road boundary candidate points, and then the candidate road boundary points are clustered to obtain a complete road boundary.

The above research mainly focuses on the extraction of boundary points of fixed regions based on single features in road geometry or spatial distribution. They have a certain improvement in detection accuracy and real-time performance. However, due to the single feature selection, the robustness of the above method is significantly reduced under the condition that the cloud density of the obstacles around the road is large or a large pitch angle is generated during the vehicle travel. In view of the above deficiencies, based on the characteristics of road geometry and point cloud distribution, this paper sets search windows of different sizes to search for road boundary points in the specified detection area, thus completing the detection of structured road boundaries. This method sets a search window of a corresponding size for each feature and performs boundary extraction according to different quantitative indicators, which can effectively avoid the occurrence of a missed detection in the detection process. In addition, the overall detection accuracy is above 95%, and it also fully meets the requirements of the intelligent vehicle algorithm in real-time.

## 2. Data Acquisition and Preprocessing

The road scene selected in this paper is a campus road network (Figure 1a), which is highly structured and covers different road lines. As shown in Figure 1b, the experimental platform consists of a modified “Tang Jun” electric car equipped with a Velodyne HDL-32E LIDAR and MV-VDM high-speed industrial digital camera.

### 2.1. Data Structure Analysis

The HDL-32E LIDAR equipped with the intelligent vehicle can scan 360° around the vehicle and generate 700,000 points of data per second. Such a large amount of data will seriously affect the real-time performance of the algorithm. Faced with such a large amount of point cloud data, we first consider the characteristics of the radar itself, determine the actual required laser beam by dynamically selecting the region of interest, and filter the excess harness (Section 2.2). On this basis, the virtual LIDAR model is established based on the coordinate system of the vehicle, and the foreground point cloud area is obtained by setting the corresponding boundary threshold (Section 2.3). After the above two steps, the amount of point cloud data can be effectively reduced, thereby saving the computation time of the subsequent algorithm.

In the process of detecting obstacles, the LIDAR data often has difficulty in clustering obstacles due to the sparse density of point clouds. The road boundary studied in this paper has less requirements on the density of the point cloud aggregation because of its own geometric features. However, when extracting and fitting boundary feature points, the algorithm is susceptible to noise. Therefore, how to accurately eliminate noise and effectively retain feature points is the key to realize road boundary detection. In this paper, a new road boundary detection algorithm is proposed for this problem. The road point cloud features are analyzed from multiple angles, and the target point cloud is accurately extracted through two closely related searches (Section 3.1 and Section 3.2).

### 2.2. Sensor Sensing Area Selection

The sensors and environmental parameters required for the experiment are shown in Table 1. The 3D LIDAR selected in this experiment has a field of view scan range of about 42° in the vertical direction. In the process of the automatic driving road perception, a part of the scanning result in the range is often redundant information, which easily interferes with the sensing result and increases the amount of calculation. Therefore, in order to maximize the utility of the algorithm, the corresponding scanning range should be selected according to the actual detection distance and the needs of the sensing object.

As shown in Figure 2, the selection of the laser radar scanning range in this experiment mainly depends on the angle *θ* between the laser beam and the horizontal line.

The angle *θ* is determined by the height of LIDAR from the ground *h*, the horizontal distance of the LIDAR from the front of the car *d*_1_, and the safety warning distance *d*_2_. According to the geometric relationship in Figure 2, the calculation formula is as follows:(1)$$\theta =\arctan\frac{h}{d_{1}+d_{2}}$$
where *d*_1_ + *d*_2_ is the LIDAR’s maximum detection distance actually processed by the algorithm, and *d*_1_ can be dynamically estimated by the FSDM (Fixed Safety Distance Model) security warning distance model [32]. The solution model can be expressed as:(2)$$d_{2}=k\times v\times t+\frac{v^{2}}{2\times j\times a}+l$$
where *k* is the driver response coefficient, which is used here as the sensor coordination response coefficient for safety reasons; *j* is the brake deceleration adjustment factor of the car; *a* is the brake deceleration of the car; and *l* is the minimum safe distance.

Part of the scanning line emitted by the LIDAR is above the horizontal line, which is not helpful for detection. According to *θ*, the number of LIDAR scan lines can be dynamically selected at different speeds (shown in Figure 3), which not only filters out unhelpful scan lines, but also effectively reduces the search range of the latter algorithm.

### 2.3. Foreground Point Cloud Area Extraction

After selecting the LIDAR scan lines according to different speeds in 2.2, we need to further determine the organizational structure of the point cloud data. Based on the virtual scan model under the polar coordinate grid [33], this paper establishes a virtual radar model based on the original point cloud. The projection of the virtual scan line of each angle of the LIDAR on the x-y plane is an extension line passing through the LIDAR rotation center and is scanned by the nearest distance. The points point to the farthest point of the scan distance, and each scan line contains 32 child nodes (the child nodes may be empty). The scan data of each frame can be expressed as a three-dimensional matrix *A_i_*:(3)$$A_{i}=\left[\begin{array}{ccccc} P_{\left(1,0\right)} & \cdots & P_{\left(1,m\right)} & \cdots & P_{\left(1,n\right)}\\ \vdots & \ddots & \vdots & \ddots & \vdots \\ P_{\left(s,0\right)} & \cdots & P_{\left(s,m\right)} & \cdots & P_{\left(s,n\right)}\\ \vdots & \ddots & \vdots & \ddots & \vdots \\ P_{\left(b,0\right)} & \cdots & P_{\left(b,m\right)} & \cdots & P_{\left(b,n\right)} \end{array}\right]$$
(4)$$P_{\left(s,m\right)}=\left[x\;y\;z\right]$$
where *P*_(*s*,*m*)_ is the attribute of the point, consisting of the three-dimensional coordinates of the point, *m* is the number of acquisitions from the starting position, *n* is the number of samples per frame, *s* is the number of laser rays, from negative to positive from 1 to 32 depending on the angle of view, *b* is the maximum value of the laser rays, which is determined by the speed of the vehicle. Each row of *A_i_* represents data acquired by a laser line rotating 360 degrees, and each column of *A_i_* represents data collected by *b* laser lines at a certain rotation angle.

The experimental platform needs to unify the LIDAR coordinate system *O*_2_*XYZ* to the vehicle coordinate system *O*_1_*XYZ*. In general, there is only rotation around the *Z* axis in the above two axes [34]. Therefore, the coordinate value of the *Z* direction is unchanged, and the LIDAR can transform the *O*_2_*XYZ* coordinate system to *O*_1_*XYZ* by rotating the matrix under the condition of rotating counterclockwise around the *Z* axis. The formula is as follows:(5)$$\left[\begin{array}{l} x_{1}\\ y_{1}\\ z_{1} \end{array}\right]=\left[\begin{array}{ccc} \cos\alpha & -\sin\alpha & 0\\ \sin\alpha & \cos\alpha & 0\\ 0 & 0 & 1 \end{array}\right]\times \left[\begin{array}{l} x_{2}\\ y_{2}\\ z_{2} \end{array}\right]+\left[\begin{array}{l} x_{0}\\ y_{0}\\ \;0 \end{array}\right]$$
where α is the angle of rotation around the *Z* axis, (*x*_1_, *y*_1_, *z*_1_) is the coordinate value of the point under the vehicle coordinate system *O*_1_*XYZ*, (*x*_2_, *y*_2_, *z*_2_) is the coordinate value of the point under the LIDAR coordinate system *O*_2_*XYZ*, and (*x*_0_, *y*_0_, 0) is the position of the LIDAR coordinate origin relative to the vehicle coordinate system.

The detection of lane boundaries is primarily based on the perception of the foreground point cloud data in the direction of travel of the vehicle. Therefore, in order to reduce the amount of data calculation, this paper sets the corresponding boundary threshold according to the actual road width and the LIDAR forward limit distance in the vehicle coordinate system, thus completing the extraction of the foreground point cloud area. The extracted foreground point clouds of the straight line and curve are shown in Figure 4a,b.

## 3. Extraction and Fitting of Road Boundary Points

The general method of the road boundary extraction is to separate the ground from the non-ground [35,36,37], filter out the ground point cloud and then cluster the analysis of the point cloud formed by the obstacle [38,39]. Based on the extraction of the foreground point cloud, this paper first extracts the boundary candidate points by using the road geometry features, and then the residual interference points are filtered two times through the spatial distribution characteristics of the point cloud, so that the accurate extraction of the road boundary is completed.

### 3.1. Geometric Feature Extraction

Structured roads have relatively regular height difference mutations at the boundary, and this feature can be used to initially extract road boundaries. First, the point cloud data is mapped to the two-dimensional plane (*x*, *y*) while retaining the height information *Z* in the vehicle reference coordinate system, before the search window size and the step size are determined according to the actual road edge width. According to the ergodic method [27], each time a search is completed, the *Z* values of all the points are arranged in descending order, and then the *n* values are selected at the front and the end of the series as the maximum and minimum reference sets, respectively. To avoid being affected by extreme values, the sample as a whole is characterized by the median in the reference set. The study found that the sample size is too small to be affected by individual mutation points; if the sample size is too large, the difference in characterization will not be significant. Therefore, the value of *n* should not be too large or too small. For the 32-line LIDAR scan results, this paper suggests that the value of *n* is within the range of 10–15% of the total number of search point clouds. The *Z* value height difference of each search window can be obtained by the following formula:(6)$$Z_{}^{\max}={\left[Z_{1},\;Z_{2},\;\ldots \;\ldots ,\;Z_{n-1},\;Z_{n}\right]}^{T}$$
(7)$$Z^{\min}={\left[Z_{N-n+1},\;Z_{N-n+2}\ldots \;\ldots ,\;Z_{N-1},\;Z_{N}\right]}^{T}$$
(8)$$R=\left\{ \begin{array}{ll} Z_{(n+1)/2}^{\max}-Z_{(n+1)/2}^{\min} & n\;is\;odd\\ \frac{Z_{\left(n/2\right)}^{\max}+Z_{\left(n/2+1\right)}^{\max}}{2}-\frac{Z_{\left(n/2\right)}^{\min}+Z_{\left(n/2+1\right)}^{\min}}{2} & n\;is\;even \end{array}\right.$$
where *Z*^max^ is a set of the first n numbers in the *z*-value series, *Z*^min^ is a set of the last n numbers in the *z*-value series, *N* is the total number of points in each search window, and *R* is used to characterize the height difference in the search window.

*R* is compared with the preset upper and lower thresholds δmin and δmax, and if *R* is within the upper and lower height difference threshold, the points in the region are reserved as candidate points for the boundary. After that, an *N* × 2 matrix is established to store the two-dimensional coordinates (*x*, *y*) of the candidate point as a candidate window. If *R* does not meet the threshold range requirement, the area is set to an empty set, that is, the points in this window are invalid points. The search process is shown in Figure 5.

The geometric feature can be used to complete the rough extraction of the road boundary points and serve as boundary candidate points. However, there is still a certain amount of noise around the road boundary points, especially when the vehicle has a pitch angle or a large slope change in the road ahead; the noise will increase, which will easily cause a large interference to the boundary fitting. In order to further improve the accuracy of the road boundary fitting, this paper uses the difference of point cloud spatial distribution to filter noise more accurately based on the candidate window obtained by the first search.

### 3.2. Spatial Distribution Feature Extraction

The LIDAR projects the scan line onto a certain shape of the clustering point cloud formed on the object, thereby realizing the external feature description of the surrounding things [40]. After a search in 3.1, high-density point cloud obstacles have been filtered out, and the spatial distribution of point clouds can be used to extract road boundaries more accurately.

The secondary search window size is 1/n of the primary window (in this paper, the secondary search window is 1/3 of a search window). We are inspired by the pixel matrix of the image to create two matrices for storing the positive and negative directions of the search results [41].The vehicle coordinate system *y*-axis is taken as the center line, and the search is performed simultaneously in the positive and negative directions of the *x*-axis. Each time the horizontal search is completed, the search result in the positive direction is recorded as a positive unit, and the search result in the negative direction is recorded as a negative unit, thereby completing the search for the candidate window in sequence. Our study calculates the point cloud density of the search window in each unit and extracts the window with the maximum density, before using the point cloud in the window as the final boundary point cloud. The point cloud in the window with the maximum density is considered as the boundary point cloud in the unit (Figure 6). The calculation equation is as follows:(9)$$Q^{+}=\left[\begin{array}{cccc} \rho_{\left(1,1\right)}^{+} & \rho_{\left(1,2\right)}^{+} & \cdots & \rho_{\left(1,j\right)}^{+}\\ \rho_{\left(2,1\right)}^{+} & \ddots & \cdots & \rho_{\left(2,j\right)}^{+}\;\\ \vdots & \vdots & \ddots & \vdots \\ \rho_{\left(i,1\right)}^{+} & \rho_{\left(i,2\right)}^{+} & \cdots & \rho_{\left(i,j\right)}^{+} \end{array}\right]\;Q^{-}=\left[\begin{array}{cccc} \rho_{\left(1,1\right)}^{-} & \rho_{\left(1,2\right)}^{-} & \cdots & \rho_{\left(1,j\right)}^{-}\\ \rho_{\left(2,1\right)}^{-} & \ddots & \cdots & \rho_{\left(2,j\right)}^{-}\\ \vdots & \vdots & \ddots & \vdots \\ \rho_{\left(i,1\right)}^{-} & \rho_{\left(i,2\right)}^{-} & \cdots & \rho_{\left(i,j\right)}^{-} \end{array}\right]$$
(10)$$\rho_{\left(i,j\right)}=\frac{\sum \left(P_{1},P_{2},\;\ldots \;,P_{n}\right)}{l\times w}$$
where *Q*^+^, *Q*^−^ are matrices for storing the positive direction search and negative direction search results, respectively. Each row in the matrix represents a search unit and records the results of a horizontal search. *ρ*^+^ and *ρ*^−^ respectively represent the point cloud spatial density of each search window in the positive and negative directions. *n* indicates the number of points. *l* indicates the length of the search window. *w* indicates the width of the search window.

After the above two different scale window searches, the interference points, such as road planes and obstacles, have been basically filtered out, so the boundary points are less affected by noise during the fitting process. Considering that the actual road line shape is generally a straight and regular curve, this paper uses the quadratic curve model [27] to fit the road boundary. The fitting model is as follows:(11)$$y=a_{0}x^{2}+a_{1}x+b$$
where *a*_0_, *a*_1_, and *b* are the undetermined coefficients of the curve fitting. When fitting a straight line, *a*_0_ is zero. 

## 4. Experiments and Results

### 4.1. Experimental Design

This study is based on the proposed structured road boundary extraction algorithm for carrying out real vehicle experiments on campus. The computing platform is configured as an Intel Core i5-8500 6-core 3.00 GHz processor with 16 GB of memory. The experimental environment is the internal road of the campus, which is characterized by a high degree of structure, and covers common road lines such as straight lines and curves. The specific experimental route is shown in Figure 7.

As shown in Figure 8, the LIDAR scanning line is sequentially scanned from the bottom to the top with a vertical resolution of 1.33°. Velodyne HDL-32E LIDAR can emit 32 scanning lines in the vertical direction. In this paper, the 21st scanning line is selected as the critical line bundle of the maximum detection distance according to the maximum speed limit of the experimental site. The scan lines above the wire harness are filtered out to obtain a LIDAR scanning area adapted to the experimental scene.

In this paper, the first search window size is 15 cm × 30 cm, the step length is 15 cm along the *X*-axis direction, and the step length is 15 cm along the *Y*-axis direction, so as to avoid the influence of individual abrupt points to ensure the continuity of the boundary. In the second search, the search window is scaled proportionally, the search window size is 5 cm × 10 cm, the step length is 2.5 cm along the *X*-axis direction, and the step length is 10 cm along the *Y*-axis direction. This ensures that all points in the candidate window are detected, and prevents the loss of real boundary points.

### 4.2. Results Evaluation

In order to evaluate the effect of our algorithm on road boundary detection, we compare it with another boundary extraction method in the experiment. The detection result is quantified by the accuracy ratio, that is, the ratio between the number of frames that successfully identify the road boundary and the actual number of acquired frames. In this paper, the detection scene is divided into straight and curved roads. The actual acquisition frames of the straight line and the curve are 1208 frames and 397 frames, respectively. The accuracy ratio under the condition of the straight line and curved road is shown in Figure 9.

Through a comparison, it can be clearly found that the traditional extraction algorithm with a single feature as reference is far lower in accuracy than our algorithm (method ③).In addition, in the straight line road detection, the height difference method has a higher detection accuracy than the point cloud density method. However, on the curved road, the result is just the opposite. 

The height difference extraction algorithm is mainly based on the difference in height between the structured road edge and the surrounding point cloud data, but the method is susceptible to the vehicle pitch angle and the slope of the road itself. Moreover, when the surrounding vegetation is dense, the result of the extraction will contain a large amount of noise. The density extraction algorithm mainly performs the extraction task according to the spatial distribution characteristics of the point cloud. If the method does not divide the ground and the non-ground, the recognition disorder will be caused by the density of the point cloud of the obstacle. On the straight line, the detection accuracy of the method ① is relatively high because the roadside environment is relatively simple and there is no large slope change. After entering the curved road environment, because the road surface has a certain slope change and the roadside vegetation is relatively tall, method ① is easy to detect incorrectly, and method ② is more suitable for the scene.

In order to further evaluate the effect of the algorithm, we randomly select 80 frames of data in the straight line and curve scene from the data that accurately identifies the road boundary. Then, the detected road width is compared with the actual road width, wherein the actual average width of the straight road is 10 m, and the actual average width of the curved road is 8 m. The comparison results are shown in Figure 10.

By comparison, the average detection error of the straight road width is 0.05 m, and the average detection error of the curved road width is 0.11 m. Compared with straight roads, curved roads have relatively large identification deviations. There are two main reasons for this: (1) After extracting the curve point cloud data, there is still a small amount of noise, which affects the fitting effect. (2) Due to the different curvature of each part of the curved road, there is a certain deviation in the determination of the fitting model parameters. The deviation is far less than the current positioning error of the civil GPS, so the algorithm can accurately complete the identification of the driving area. The effect of the algorithm in the different stages of boundary extraction of the straight line and curve is shown in Figure 11.

It can be seen from the effect of the road boundary extraction that the algorithm can accurately detect the structured road boundaries with different linear shapes, especially for the detection of the straight line. A small amount of noise will appear in the extraction of the curve boundary points. The main reasons are as follows: (1) There is a deviation angle between the search direction and the road linear trend, and some noise is in the search window of the real boundary. (2) The height difference between the obstacle and the ground, and the spatial distribution of the point cloud itself satisfy the above two boundary search conditions.

It has been verified by actual road tests that the noise generated by the above two cases rarely occurs and has little effect on the final road boundary fitting. The processing time of each frame in the actual road detection in this paper is shown in Figure 12. The average processing speed is 41.8 ms/frame, which fully meets the real-time requirements of an intelligent vehicle.

After evaluating the accuracy and real-time of the algorithm, this paper further explores the robustness of the algorithm in complex scenarios. We select three representative complex scenes on campus and conduct a qualitative analysis of the test results.

Figure 13 shows the most common relatively complex traffic conditions (Scene 1). As shown in Figure 13a, there are obstacles such as electric vehicles and pedestrians in the road, which easily block the road boundary. Moreover, the right side of the road is a relatively low lawn, and it is difficult to separate the road boundary by the height difference method only. The road boundary can be detected more stably by the method in this paper (Figure 13b), mainly because the other disturbance points are filtered according to the spatial distribution characteristics of the road boundary while eliminating the non-ground points. It can be seen from the fitting results (Figure 13c) that the proposed algorithm is robust to obstacles on roads and roadsides.

Scene 2 is a typical T-shaped intersection. As shown in Figure 14a, the T-shaped intersection will cause one side of the road boundary to be interrupted. This situation does not affect the algorithm’s identification of the road boundary (Figure 14b), because the algorithm extracts the boundary in accordance with the characteristics of the point cloud rather than the road structure. However, since the point cloud distribution is multi-segmented, the fitting model can only express the expression of two sides on both sides at most, so some partial boundaries are not fitted (Figure 14c). 

Scene 3 is a typical Y-shaped intersection. The Y-shaped intersection is similar to the cross-shaped intersection, as shown in Figure 15a, and its boundary line shape is mainly curved. Through the algorithm, the non-road boundary points can be successfully filtered out (Figure 15b), and the fitting situation is similar to the T-shaped intersection (Figure 15c).

We select the first 1500 frames of data as the data set, and use the results of the manual labeling as a benchmark. Then, the proposed method is compared to the results obtained by other curb detection algorithms (Table 2). 

In the average period, Hata et al. proposed the fastest algorithm because the algorithm uses a point grid to organize point cloud data, which reduces the average processing speed of the point cloud. The speed of this algorithm is second only to [24], faster than the other two algorithms, and fully meets the requirements of the automatic driving real-time. In terms of the precision rate, Zhang et al. only used the method of height difference in the scan line but did not consider the obstacle on the road, so the accuracy was relatively low. Liu et al. and Hata et al. also tested only on the basis of the single local feature of the road. However, the method proposed in this paper not only considers the obstacles on the road and the surroundings, but also uses a combination of various local features to detect the boundary points, so the accuracy is higher than other algorithms. The average width error is an important evaluation index that reflects the accuracy of the algorithm extraction. It refers to the absolute value of the difference between the road width after detection and the actual road width. In order to better distinguish between boundary points and non-boundary points, our algorithm uses a twi-extraction method, so that the noise around the boundary can be filtered out. Oppositely, the other three algorithms were not thorough enough to filter the edge noise, which caused the determination of the boundary candidate points to deviate.

## 5. Conclusions

This paper proposes a structured road boundary detection algorithm based on a multi-feature combination. First, in the complex point cloud data, the foreground point cloud area is extracted according to the road environment and the characteristics of the LIDAR. Then, we use the road geometry and the point cloud spatial distribution feature to perform the first and second search respectively. Finally, the road boundary is fitted by a quadratic model. The experimental results show that the detection accuracy of the algorithm on the straight line is over 97%, and that the detection accuracy on the curve is over 92%. In addition, after two search results, the multi-feature extraction algorithm in this paper has a greater improvement in accuracy than the single feature extraction. The average detection period of the algorithm is 41.8 ms, which fully meets the real-time requirements of the intelligent vehicle. Finally, the algorithm is further verified in different complex scenarios, and the recognition effect is still stable.

The algorithm has a strong robustness to the detection of road boundaries when the vehicle has a pitch angle and a large gradient in front of the road. It can solve the problems of more noise and the irregular extraction of boundary line shape under single feature extraction. However, the accuracy of the algorithm in the detection of curve boundaries needs to be further improved. In the future, the search strategy in the algorithm will continue to be optimized for the above deficiencies, and the next step of unstructured road boundary detection will be studied.

## Figures and Tables

**Figure 1 sensors-19-01551-f001:**
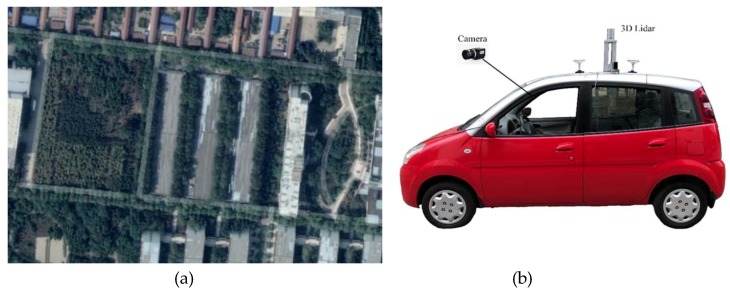
Experiment environment and platform of automatic driving. (**a**) is the actual road scene of the experiment. (**b**) is our experimental platform for algorithm verification.

**Figure 2 sensors-19-01551-f002:**
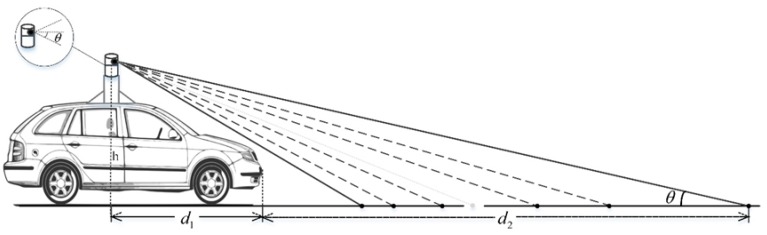
Schematic of 3D LIDAR vertical scanning range.

**Figure 3 sensors-19-01551-f003:**
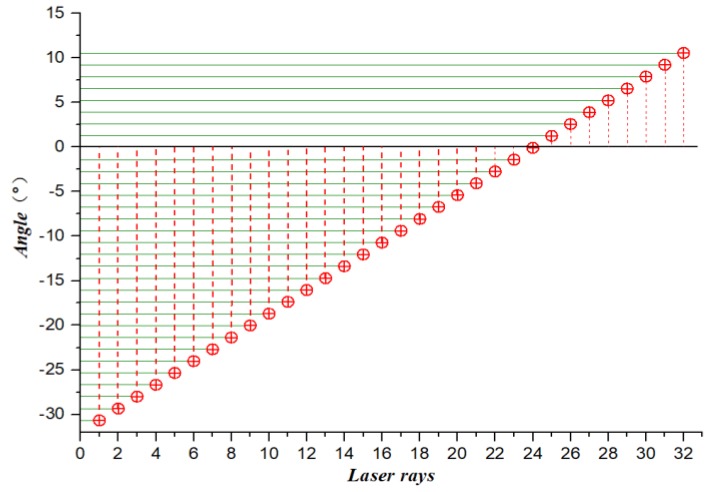
Schematic of 3D LIDAR scan lines selection.

**Figure 4 sensors-19-01551-f004:**
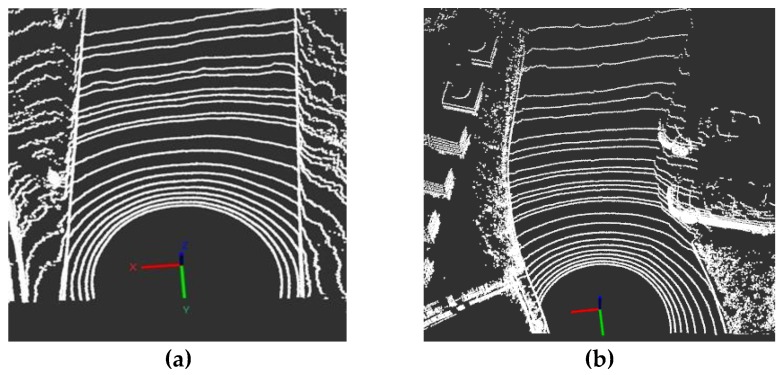
Extraction of foreground point cloud. (**a**) is the extracted foreground point clouds of the straight line. (**b**) is the extracted foreground point clouds of the curve.

**Figure 5 sensors-19-01551-f005:**
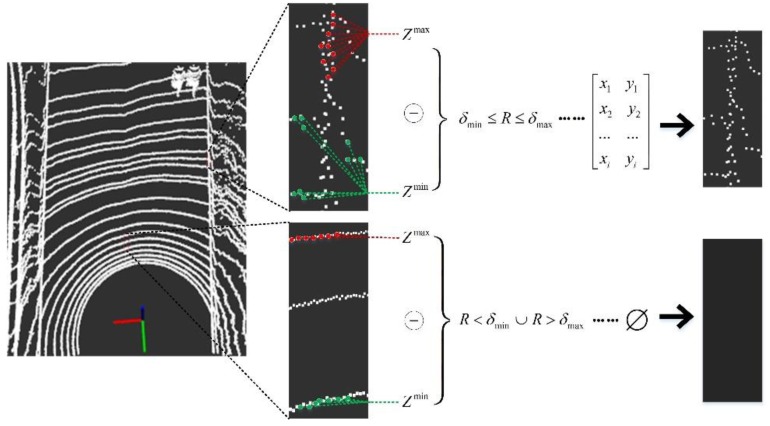
Schematic of the first search window. In this search, we propose the concept of the largest cluster point set (*Z*^max^) and the smallest cluster point set (*Z*^min^).

**Figure 6 sensors-19-01551-f006:**
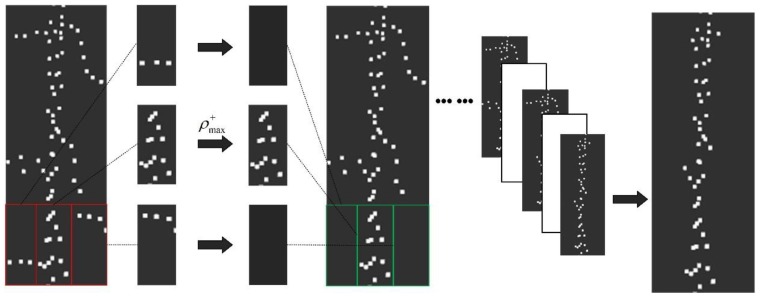
Schematic of the second search window.

**Figure 7 sensors-19-01551-f007:**
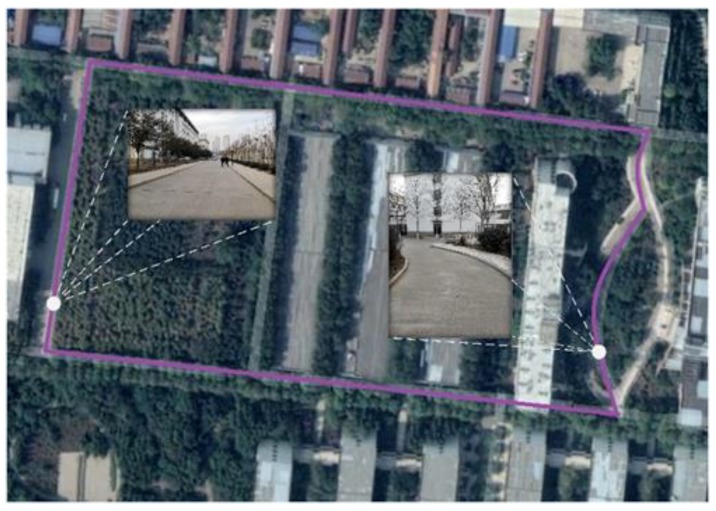
Schematic of the experimental route.

**Figure 8 sensors-19-01551-f008:**
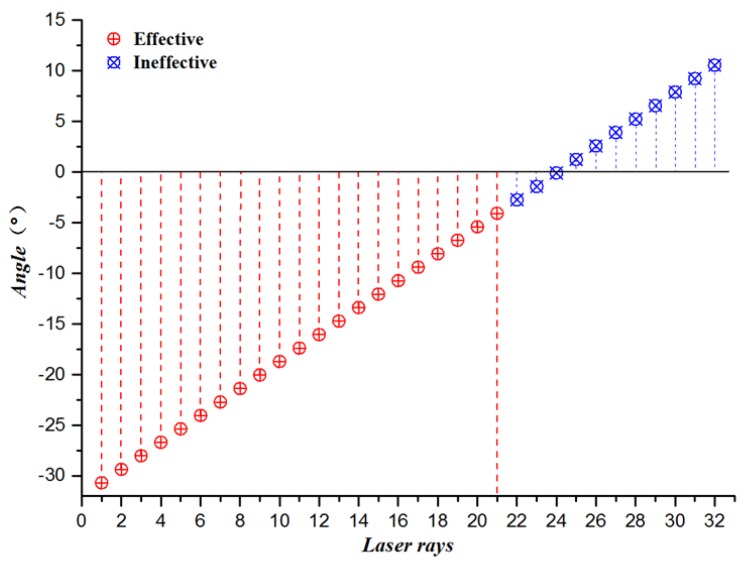
Schematic of the 3D Lidar scanning range selection.

**Figure 9 sensors-19-01551-f009:**
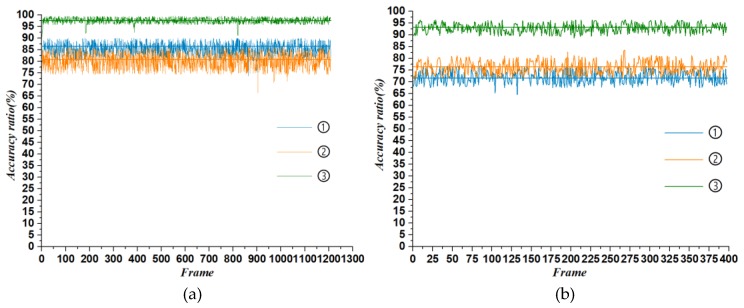
The accuracy ratio of different methods. (**a**) represents the accuracy ratio of the straight line road, and (**b**) represents the accuracy ratio of the curved road. In the figure: ① is the high difference extraction method, ② is the point cloud density extraction method, and ③ is the application method of this paper.

**Figure 10 sensors-19-01551-f010:**
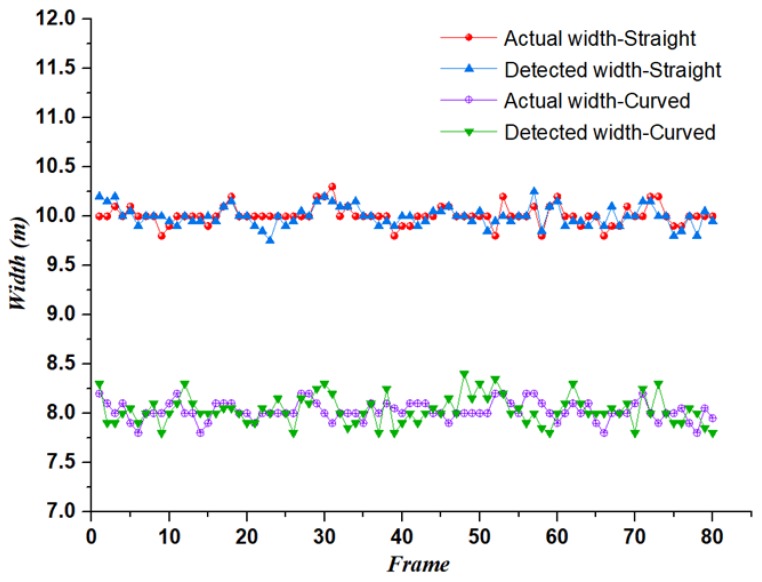
Comparison of detected road width and actual road width.

**Figure 11 sensors-19-01551-f011:**
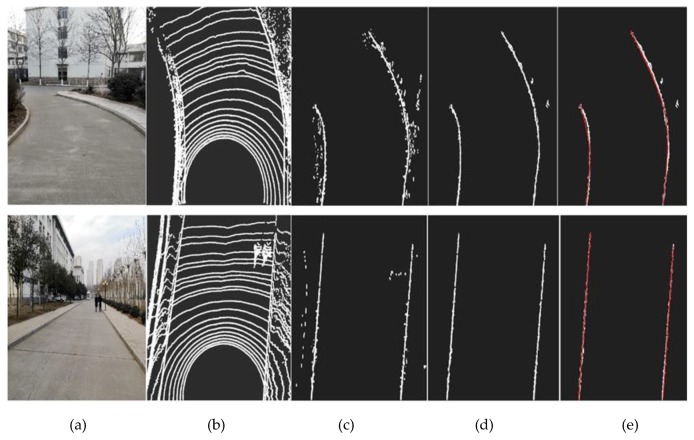
The effect of road boundary extraction, (**a**) represents actual roads collected for the camera, (**b**) represents the extraction result of the foreground point cloud area, (**c**) represents the effect the first search, (**d**) is the effect of the second search, and (**e**) is the final road boundary fitting effect.

**Figure 12 sensors-19-01551-f012:**
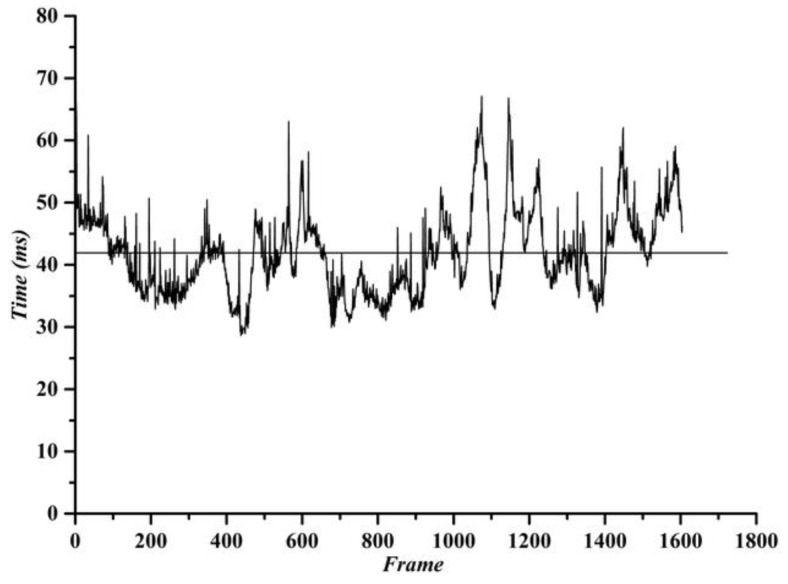
Time of algorithm running.

**Figure 13 sensors-19-01551-f013:**
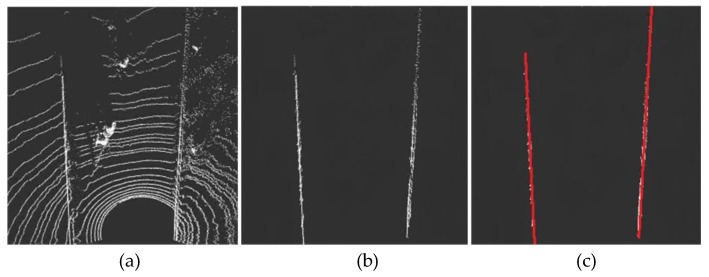
Curb detection under a complex road scene. (**a**) is the extracted foreground point clouds of road where there are obstacles. (**b**) is the boundary result extracted by the algorithm. (**c**) is the boundary fitting result.

**Figure 14 sensors-19-01551-f014:**
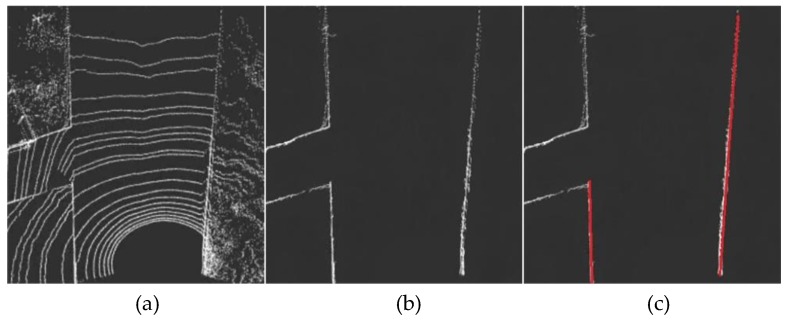
Curb detection of a typical T-shaped intersection scene. (**a**) is the extracted foreground point clouds of T-shaped intersection. (**b**) is the boundary result extracted by the algorithm. (**c**) is the boundary fitting result.

**Figure 15 sensors-19-01551-f015:**
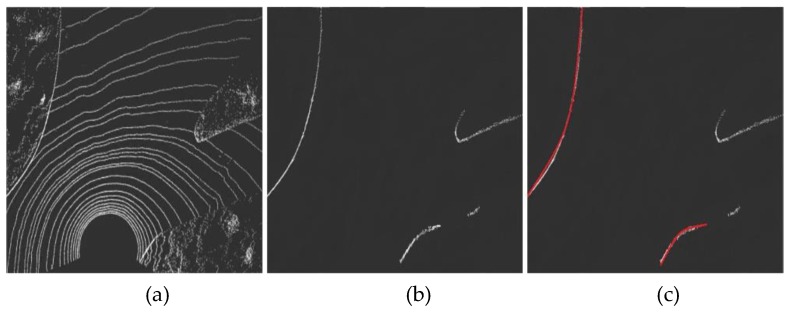
Curb detection of a typical Y-shaped intersection scene. (**a**) is the extracted foreground point clouds of Y-shaped intersection. (**b**) is the boundary result extracted by the algorithm. (**c**) is the boundary fitting result.

**Table 1 sensors-19-01551-t001:** Parameters of experimental platform and environment.

Experimental Object	Parameter Index	Parameter Value
Intelligent vehicle	*h*	2.2 m
	*d* _1_	2.4 m
LIDAR	Horizontal field of view	360°
	Vertical field of view	−30.67~10.67°
	Vertical resolution	1.33°
	Scanning frequency	10 Hz
	Maximum detection distance	70 m
Camera	Maximum resolution	1280 * 960

Note: “*h*” indicates the height of LIDAR from the ground; and “*d*_1_” indicates the horizontal distance of the LIDAR from the front of the car.

**Table 2 sensors-19-01551-t002:** The comparison between different curb detection algorithms.

Methods	Average Period/ms	Precision Rate/%	Average Width Error/m
Proposed	**41.8**	**96.47**	**0.08**
Hata et al. [24]	27.4	90.38	0.15
Liu et al. [27]	46.1	92.40	0.14
Zhang et al. [30]	63.4	86.79	0.23

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
