# Peer review of "A Multi-Feature Search Window Method for Road Boundary Detection Based on LIDAR Data"

_sensors, 2019, doi:10.3390/s19071551_

Round 1

Reviewer 1 Report

This manuscript presented a method to extract road boundary from point cloud data LIDAR. In order to improve the accuracy of structured road boundary a search window methods is built.

1) In the introduction on the research background, the road boundary extraction from other data source, such as GPS trajectory data, also need to be summarized using asocial reference manuscripts.

2) The difficulties in the road boundary extraction by LIDAR data  needs to be discussed ,especially for some key questions in the point cloud data process. How to settle the large volume data question?

3) As for the context for different road landscape, the extraction method needs to explore the aiming measures, for example the road joint of cross shape, or triangle shape. As for the complex scene in neighborhood analysis around the road, the method needs to give deep discussion.

4) the accuracy evaluation needs to present deep discussion rather than giving a table.    

Author Response

Response to Reviewer 1 Comments

Dear Editors and Reviewers:

Thank you for your letter and for the reviewers' comments concerning our manuscript entitled '' A Multi-Feature Search Window Method for Road Boundary Detection Based on LIDAR Data '' (ID: sensors-460531). Those comments are all valuable and very helpful for revising and improving our paper, as well as the important guiding significance to our researches. We have studied comments carefully and have made correction which we hope meet with approval. Revised portion are marked in red in the paper. The main corrections in the paper and the responds to the reviewer's comments are as flowing.

Responds to the reviewer’s comments:

Point 1: In the introduction on the research background, the road boundary extraction from other data source, such as GPS trajectory data, also need to be summarized using asocial reference manuscripts.

Response 1: We have further referred to other reference manuscripts to improve the introduction according to the Reviewer's suggestion. As follows:

The GPS is an indispensable technology for current driving positioning and plays a very important role in autonomous vehicles positioning. In the research on autonomous vehicles, Wu et al. [1] design a real-time autonomous navigation system by integrating a real-time kinematics differential global position system (RTK-DGPS). The main destination of AUTOPIA program [2, 3] in Europe is to develop autonomous vehicles by using GPS sensors for navigation and fuzzy logic for lateral control. Lundgren et al. [4] presented a localization algorithm based on a map and a set of off-the-shelf sensors, with the purpose of evaluating a low-cost solution with respect to localization performance. In addition, GPS data is also essential to improve map information. For example, in order to enrich the information of OpenStreetMap, Mobasheri et al. [5] constructed sidewalk geometries by mining data from GPS data of people in wheelchairs.

Rami et al. [13] proposed a novel detection algorithm for vision systems based on combined fuzzy image processing and bacterial algorithm, which has been modified and applied to produce time-based trajectory for the optimal path. Seibert et al. [14] described the extension of an existing commercial lane detection system for marked roads by the detection of soft shoulders, curbs and guardrails based on a front directed gray scale camera. Zhang et al. [15] used a tracking camera to propose a local adaptive threshold image binary segmentation algorithm for path recognition, and then accurately extracted the strip centerline for path tracking to accurately complete the unmanned vehicle path tracking. Oniga and Nedevschi [16] proposed a real-time algorithm for curb detection based on dense stereovision, and transformed the 3D points from stereovision into a Digital Elevation Map.

Nan et al. [18] developed a road boundary detection algorithm to extract road features as line segments in polar coordinates relative to the 2D Lidar sensor. In order to creating map based on Lidar in GPS-denied environments, Kang et al. [21] proposed a high precision 2D laser point-clouds map creating method based on loop closure detection. Liu et al. [22] used 2D sequential laser range finder data and vehicle state information to build a local Digital Elevation Map (DEM), and achieved curb detection with 1D Gaussian process regression.

Point 2: The difficulties in the road boundary extraction by LIDAR data needs to be discussed, especially for some key questions in the point cloud data process. How to settle the large volume data question?

Response 2: Considering the Reviewer's suggestion, we have specifically added a section to discuss some key issues in the point cloud data process and as follows:

2.1. Data structure analysis

The HDL-32E LIDAR equipped with the intelligent vehicle can scan 360° around the vehicle and generate 700,000 points of data per second. Such a large amount of data will seriously affect the real-time performance of the algorithm. Faced with such a large amount of point cloud data, we first consider the characteristics of the radar itself, determine the actual required laser beam by dynamically selecting the region of interest, and filter the excess harness (Section 2.2). On this basis, the virtual LIDAR model is established based on the coordinate system of the vehicle and the foreground point cloud area is obtained by setting the corresponding boundary threshold (Section 2.3). After the above two steps, the amount of point cloud data can be effectively reduced, thereby saving the computation time of the subsequent algorithm.

In the process of detecting obstacles, the LIDAR data often has difficulty in clustering obstacles due to the sparse density of point clouds. The road boundary studied in this paper has less requirements on the density of point cloud aggregation because of its own geometric features. However, when extracting and fitting boundary feature points, the algorithm is susceptible to noise. Therefore, how to accurately eliminate noise and effectively retain feature points is the key to realize road boundary detection. In this paper, a new road boundary detection algorithm is proposed for this problem. The road point cloud features are analyzed from multiple angles, and the target point cloud is accurately extracted through two closely related searches (Section 3.1, 3.2).

Point 3: As for the context for different road landscape, the extraction method needs to explore the aiming measures, for example the road joint of cross shape, or triangle shape. As for the complex scene in neighborhood analysis around the road, the method needs to give deep discussion.

Response 3: As Reviewer suggested, we have further refined the experimental design and selected different typical complex scenarios to test the algorithm. The new results are as follows:

After evaluating the accuracy and real-time of the algorithm, this paper further explores the robustness of the algorithm in complex scenarios. We select three representative complex scenes on campus and conduct qualitative analysis of the test results.

Figure 13 shows the most common relatively complex traffic conditions (Scene 1). As shown in Figure 13(a), there are obstacles such as electric vehicles and pedestrians in the road, which easily block the road boundary. Moreover, the right side of the road is a relatively low lawn, and it is difficult to separate the road boundary only by the height difference method. The road boundary can be detected more stably by the method in this paper (Figure 13(b)), mainly because the other disturbance points are filtered according to the spatial distribution characteristics of the road boundary while eliminating the non-ground points. It can be seen from the fitting results (Figure 13(c)) that the proposed algorithm is robust to obstacles on roads and roadsides.

Scene 2 is a typical T-shaped intersection. As shown in Figure 14(a), the T-shaped intersection will cause one side of the road boundary to be interrupted. This situation does not affect the algorithm's identification of the road boundary (Figure 14(b)), because the algorithm extracts the boundary in accordance with the characteristics of the point cloud rather than the road structure. However, since the point cloud distribution is multi-segmented, the fitting model can only express the expression of at most two sides on both sides, so some partial boundaries are not fitted (Figure 14(c)).

Scene 3 is a typical Y-shaped intersection. The Y-shaped intersection is similar to the cross-shaped intersection, as shown in Figure 15(a), and its boundary line shape is mainly curved. Through the algorithm, the non-road boundary points can be successfully filtered out (Figure 15(b)), and the fitting situation is similar to the T-shaped intersection (Figure 15(c)).

Point 4: the accuracy evaluation needs to present deep discussion rather than giving a table.

Response 4: Considering the Reviewer's suggestion, we have expanded the contents of Table and discussed on accuracy evaluation deeply. As follows:

The height difference extraction algorithm is mainly based on the difference in height between the structured road edge and the surrounding point cloud data, but the method is susceptible to the vehicle pitch angle and the slope of the road itself. Moreover, when the surrounding vegetation is dense, the result of the extraction will contain a large amount of noise. The density extraction algorithm mainly performs the extraction task according to the spatial distribution characteristics of the point cloud. If the method does not divide the ground and the non-ground, the recognition disorder will be caused by the density of the point cloud of the obstacle. On the straight line, the detection accuracy of the method 1 is relatively high because the roadside environment is relatively simple and there is no large slope change. After entering the curved road environment, because the road surface has a certain slope change and the roadside vegetation is relatively tall, the method 1 is easy to detect incorrectly, and the method 2 is more suitable for the scene. (“Further discussion and analysis of Figure 9 in the original manuscript.”)

In order to further evaluate the effect of the algorithm, we randomly select 80 frames of data in the straight line and curve scene from the data that accurately identifies the road boundary. Then, the detected road width is compared with the actual road width, wherein the actual average width of the straight road is 10 m, and the actual average width of the curved road is 8 m. The comparison results are shown in Figure 10.

By comparison, the average detection error of straight road width is 0.05m, and the average detection error of curve road width is 0.11m. Compared with straight roads, curved roads have relatively large identification deviations. There are two main reasons: (1) After extracting the curve point cloud data, there is still a small amount of noise, which affects the fitting effect. (2) Due to the different curvature of each part of the curved road, there is a certain deviation in the determination of the fitting model parameters. The deviation is far less than the current positioning error of civil GPS, so the algorithm can accurately complete the identification of the driving area.

We select the first 1500 frames of data as the data set, and use the results of the manual labeling as a benchmark. Then, the proposed method is compared to the results obtained by other curb detection algorithms (Table 2).

Table 2. The comparison between different curb detection algorithms.

Methods

Average period/ ms

Precision rate/ %

Average width error/ m

Proposed

41.8

96.47

0.08

Hata et al. [24]

27.4

90.38

0.15

Liu et al. [27]

46.1

92.40

0.14

Zhang et al. [30]

63.4

86.79

0.23

In the average period, Hata et al. proposed the fastest algorithm because the algorithm uses point grid to organize point cloud data, which reduces the average processing speed of the point cloud. The speed of this algorithm is second only to the reference [24], faster than the other two algorithms, and fully meets the requirements of automatic driving real-time. In terms of precision rate, Zhang et al. only used the method of height difference in the scan line but did not consider the obstacle on the road, so the accuracy was relatively low. Liu et al. and Hata et al. also tested only on the basis of single local feature of the road. However, the method proposed in this paper not only considers the obstacles on the road and the surrounding, but also uses a combination of various local features to detect the boundary points, so the accuracy is higher than other algorithms. The average width error is an important evaluation index that reflects the accuracy of the algorithm extraction. It refers to the absolute value of the difference between the road width after detection and the actual road width. In order to better distinguish between boundary points and non-boundary points, our algorithm uses a twi-extraction method, so that the noise around the boundary can be filtered out. Oppositely, the other three algorithms were not thorough enough to filter the edge noise, which caused the determination of the boundary candidate points to deviate.

We tried our best to improve the manuscript and made some changes in the manuscript. These changes will not influence the content and framework of the paper. And here we did not list the changes but marked in red in revised paper.

We appreciate for Editors/Reviewers' warm work earnestly, and hope that the correction will meet with approval.

Once again, thank you very much for your comments and suggestions.

Reviewer 2 Report

This is an interesting article and I enjoyed reading the work. Below are some suggestions that can help you to improve the paper.

- It would be better to extend the introduction or related work sections by providing examples of similar studies that have used other methods or data (e.g. gps) for similar purpose. I understand that you have somehow done this but it would be helpful to extend this with more info and reference because this will show the importance of such objectives in urban transportation. A recent study that comes to my mind is: Enrichment of openstreetmap data completeness with sidewalk geometries using data mining techniques. Sensors18(2), 509 (which deals with processing gps trajectories for extracting sidewalk geometries). and also this study: Intelligent traffic management system for cross section of roads using computer vision. In 2017 IEEE 7th Annual Computing and Communication Workshop and Conference (CCWC) (pp. 1-7). IEEE. You can find more studies by searching the literature.

- Please make sure that reference is given for all formula's used in the study.

- The mapmatching method used in the work needs more explanations in a way that if somebody aims to perform this analysis they can read your paper and clearly perform the same study/method.

- The paper lacks a very important section: Evaluation. You need to evaluate your method and assess the quality of derived information in a proper scientific way. The current version of the work only has a short and bfrief accuracy asssessment. This needs to be properly extended and preferably have a seperate sub-section. Can you for example, compare it to other studies which detect road boundary, and see how good are your results compared to those?

Author Response

Response to Reviewer 2 Comments

Dear Editors and Reviewers:

Thank you for your letter and for the reviewers' comments concerning our manuscript entitled '' A Multi-Feature Search Window Method for Road Boundary Detection Based on LIDAR Data '' (ID: sensors-460531). Those comments are all valuable and very helpful for revising and improving our paper, as well as the important guiding significance to our studies. We have studied comments carefully and have made correction which we hope meet with approval. Revised portion are marked in red in the paper. The main corrections in the paper and the responds to the reviewer's comments are as flowing. Please refer to the uploaded PDF version for the relevant Figures in the reply.

Responds to the reviewer’s comments:

Point 1: It would be better to extend the introduction or related work sections by providing examples of similar studies that have used other methods or data (e.g. gps) for similar purpose. You can find more studies by searching the literature.

Response 1: We have further referred to other reference manuscripts to improve the introduction according to the Reviewer's suggestion. Moreover, we feel that your study is very meaningful and we will also use vehicle sensors to conduct a series of studies on the construction of high-precision maps in the near future. The supplementary content is as follows:

The GPS is an indispensable technology for current driving positioning and plays a very important role in autonomous vehicles positioning. In the research on autonomous vehicles, Wu et al. [1] design a real-time autonomous navigation system by integrating a real-time kinematics differential global position system (RTK-DGPS). The main destination of AUTOPIA program [2, 3] in Europe is to develop autonomous vehicles by using GPS sensors for navigation and fuzzy logic for lateral control. Lundgren et al. [4] presented a localization algorithm based on a map and a set of off-the-shelf sensors, with the purpose of evaluating a low-cost solution with respect to localization performance. In addition, GPS data is also essential to improve map information. For example, in order to enrich the information of OpenStreetMap, Mobasheri et al. [5] constructed sidewalk geometries by mining data from GPS data of people in wheelchairs.

Rami et al. [13] proposed a novel detection algorithm for vision systems based on combined fuzzy image processing and bacterial algorithm, which has been modified and applied to produce time-based trajectory for the optimal path. Seibert et al. [14] described the extension of an existing commercial lane detection system for marked roads by the detection of soft shoulders, curbs and guardrails based on a front directed gray scale camera. Zhang et al. [15] used a tracking camera to propose a local adaptive threshold image binary segmentation algorithm for path recognition, and then accurately extracted the strip centerline for path tracking to accurately complete the unmanned vehicle path tracking. Oniga and Nedevschi [16] proposed a real-time algorithm for curb detection based on dense stereovision, and transformed the 3D points from stereovision into a Digital Elevation Map.

Nan et al. [18] developed a road boundary detection algorithm to extract road features as line segments in polar coordinates relative to the 2D Lidar sensor. In order to creating map based on Lidar in GPS-denied environments, Kang et al. [21] proposed a high precision 2D laser point-clouds map creating method based on loop closure detection. Liu et al. [22] used 2D sequential laser range finder data and vehicle state information to build a local Digital Elevation Map (DEM), and achieved curb detection with 1D Gaussian process regression.

Point 2: Please make sure that reference is given for all formula's used in the study.

Response 2: Considering the Reviewer's suggestion, we have already noted all the formulas in the manuscript as reference sources and displayed them in red font. The formulas that have not been referenced before and their sources are as follows:

Formula (1) is derived from the geometric relationship shown in Figure 2 of the manuscript.

The reference for formula (3), (4) is explained as follows: “After selecting the LIDAR scan lines according to different speeds in 2.2, we need to further determine the organizational structure of point cloud data. Based on the virtual scan model under the polar coordinate grid [33], this paper establishes a virtual radar model based on the original point cloud. The projection of the virtual scan line of each angle of the LIDAR on the x-y plane is an extension line passing through the LIDAR rotation center and is scanned by the nearest distance. The points point to the farthest point of the scan distance, and each scan line contains 32 child nodes (child nodes may be empty).”

Formula (6), (7) and (8) are derived from the ergodic method of reference [27].

Formula (9) and (10) are based on reference [41]. “We are inspired by the pixel matrix of the image to create two matrices for storing the positive and negative directions of the search results [41].”

Point 3: The map matching method used in the work needs more explanations in a way that if somebody aims to perform this analysis they can read your paper and clearly perform the same study/method.

Response: In order to explain our algorithm more vividly and concretely, we added Figure 6 to the original algorithm description, so that the reader can understand more clearly.

I am sorry that we did not use map matching methods in the current work, because our perception of the environment is to obtain real-time data through on-board sensors, and then to visualize and process the data in the Robot Operating System (ROS) of Ubuntu16.04. ROS that we use is capable of multi-threaded parallel operation in a multi-sensor state and uses the node subscription feature to edit the received topic. After we obtain the data generated by the LIDAR, we can directly convert it into a point cloud map through the Point Cloud Library (PCL) in ROS. Then, we write our algorithm in C++ to a node under the subscription theme to achieve real-time processing of point cloud data.

Although we did not use the map matching method in the current research, we will study the related methods such as using sensors to establish high-precision maps in the later work, Simultaneous Localization and Mapping (SLAM).

Point 4: The paper lacks a very important section: Evaluation. You need to evaluate your method and assess the quality of derived information in a proper scientific way. The current version of the work only has a short and brief accuracy assessment. This needs to be properly extended and preferably have a seperate sub-section. Can you for example, compare it to other studies which detect road boundary, and see how good are your results compared to those?

Response 4: Considering the Reviewer's suggestion, we have added a separate sub-section (4.2 Results evaluation) to evaluate the algorithm, where the evaluation of accuracy adds more considerations. At the end of the paper, we compare the algorithm of this paper with other algorithms for detecting road boundaries, and obtain satisfactory results.

The height difference extraction algorithm is mainly based on the difference in height between the structured road edge and the surrounding point cloud data, but the method is susceptible to the vehicle pitch angle and the slope of the road itself. Moreover, when the surrounding vegetation is dense, the result of the extraction will contain a large amount of noise. The density extraction algorithm mainly performs the extraction task according to the spatial distribution characteristics of the point cloud. If the method does not divide the ground and the non-ground, the recognition disorder will be caused by the density of the point cloud of the obstacle. On the straight line, the detection accuracy of the method 1 is relatively high because the roadside environment is relatively simple and there is no large slope change. After entering the curved road environment, because the road surface has a certain slope change and the roadside vegetation is relatively tall, the method 1 is easy to detect incorrectly, and the method 2 is more suitable for the scene. (“Further discussion and analysis of Figure 9 in the original manuscript.”)

In order to further evaluate the effect of the algorithm, we randomly select 80 frames of data in the straight line and curve scene from the data that accurately identifies the road boundary. Then, the detected road width is compared with the actual road width, wherein the actual average width of the straight road is 10 m, and the actual average width of the curved road is 8 m. The comparison results are shown in Figure 10.

By comparison, the average detection error of straight road width is 0.05m, and the average detection error of curve road width is 0.11m. Compared with straight roads, curved roads have relatively large identification deviations. There are two main reasons: (1) After extracting the curve point cloud data, there is still a small amount of noise, which affects the fitting effect. (2) Due to the different curvature of each part of the curved road, there is a certain deviation in the determination of the fitting model parameters. The deviation is far less than the current positioning error of civil GPS, so the algorithm can accurately complete the identification of the driving area.

After evaluating the accuracy and real-time of the algorithm, this paper further explores the robustness of the algorithm in complex scenarios. We select three representative complex scenes on campus and conduct qualitative analysis of the test results.

Figure 13 shows the most common relatively complex traffic conditions (Scene 1). As shown in Figure 13(a), there are obstacles such as electric vehicles and pedestrians in the road, which easily block the road boundary. Moreover, the right side of the road is a relatively low lawn, and it is difficult to separate the road boundary only by the height difference method. The road boundary can be detected more stably by the method in this paper (Figure 13(b)), mainly because the other disturbance points are filtered according to the spatial distribution characteristics of the road boundary while eliminating the non-ground points. It can be seen from the fitting results (Figure 13(c)) that the proposed algorithm is robust to obstacles on roads and roadsides.

Scene 2 is a typical T-shaped intersection. As shown in Figure 14(a), the T-shaped intersection will cause one side of the road boundary to be interrupted. This situation does not affect the algorithm's identification of the road boundary (Figure 14(b)), because the algorithm extracts the boundary in accordance with the characteristics of the point cloud rather than the road structure. However, since the point cloud distribution is multi-segmented, the fitting model can only express the expression of at most two sides on both sides, so some partial boundaries are not fitted (Figure 14(c)).

       Scene 3 is a typical Y-shaped intersection. The Y-shaped intersection is similar to the cross-shaped intersection, as shown in Figure 15(a), and its boundary line shape is mainly curved. Through the algorithm, the non-road boundary points can be successfully filtered out (Figure 15(b)), and the fitting situation is similar to the T-shaped intersection (Figure 15(c)).

We select the first 1500 frames of data as the data set, and use the results of the manual labeling as a benchmark. Then, the proposed method is compared to the results obtained by other curb detection algorithms (Table 2).

Table 2. The comparison between different curb detection algorithms.

Methods

Average period/ ms

Precision rate/ %

Average width error/ m

Proposed

41.8

96.47

0.08

Hata et al. [24]

27.4

90.38

0.15

Liu et al. [27]

46.1

92.40

0.14

Zhang et al. [30]

63.4

86.79

0.23

In the average period, Hata et al. proposed the fastest algorithm because the algorithm uses point grid to organize point cloud data, which reduces the average processing speed of the point cloud. The speed of this algorithm is second only to the reference [24], faster than the other two algorithms, and fully meets the requirements of automatic driving real-time. In terms of precision rate, Zhang et al. only used the method of height difference in the scan line but did not consider the obstacle on the road, so the accuracy was relatively low. Liu et al. and Hata et al. also tested only on the basis of single local feature of the road. However, the method proposed in this paper not only considers the obstacles on the road and the surrounding, but also uses a combination of various local features to detect the boundary points, so the accuracy is higher than other algorithms. The average width error is an important evaluation index that reflects the accuracy of the algorithm extraction. It refers to the absolute value of the difference between the road width after detection and the actual road width. In order to better distinguish between boundary points and non-boundary points, our algorithm uses a twi-extraction method, so that the noise around the boundary can be filtered out. Oppositely, the other three algorithms were not thorough enough to filter the edge noise, which caused the determination of the boundary candidate points to deviate.

We tried our best to improve the manuscript and made some changes in the manuscript. These changes will not influence the content and framework of the paper. And here we did not list the changes but marked in red in revised paper.

We appreciate for Editors/Reviewers' warm work earnestly, and hope that the correction will meet with approval.

Once again, thank you very much for your comments and suggestions.

Reviewer 3 Report

the paper presents a method for road boundary extraction that is based on a custom designed feature which seems to be easy and effective. This work goes against the current research streamline of doing it by learning. 

The paper is fairly well written and easy to read, some small adjustments are required. 

The results section is satisfactory it contains a comparison with another method and it shows the accuracy of the current detection over 97%. 

The authors argue that the work is motivated by the fact that current research is done using cameras and 2d lasers, however this is questionable according to many recent paper fusing cameras and lidars to improve detection.

Author Response

Response to Reviewer 3 Comments

Dear Editors and Reviewers:

Thank you for your letter and for the reviewers' comments concerning our manuscript entitled '' A Multi-Feature Search Window Method for Road Boundary Detection Based on LIDAR Data '' (ID: sensors-460531). Those comments are all valuable and very helpful for revising and improving our paper, as well as the important guiding significance to our studies. We have studied comments carefully and have made correction which we hope meet with approval. Revised portion are marked in red in the paper. The main corrections in the paper and the responds to the reviewer's comments are as flowing:

Responds to the reviewer’s comments:

Point 1: the paper presents a method for road boundary extraction that is based on a custom designed feature which seems to be easy and effective. This work goes against the current research streamline of doing it by learning.

Response 1: Thanks for your kind words.

Point 2: The paper is fairly well written and easy to read, some small adjustments are required.

Response 2: Thank you a lot and We have improved the inadequacies of the content of the article.

Point 3: The results section is satisfactory it contains a comparison with another method and it shows the accuracy of the current detection over 97%. 

Response 3: Thank you for your suggestion, in order to further evaluate our algorithm, we have further increased our evaluation indicators.

Point 4: The authors argue that the work is motivated by the fact that current research is done using cameras and 2d lasers, however this is questionable according to many recent paper fusing cameras and lidars to improve detection.

Response 4: Your suggestion is very valuable. After reading a large number of references, we find that the algorithm for extracting road boundaries mainly relies on camera or laser radar, and the detection algorithm for camera and LIDAR fusion is mainly for the detection of obstacles. The reason for not referring to the article on sensor fusion in this manuscript is as follows:

Processing each sensor data is mainly divided into two steps, segmentation and classification. LIDAR's long detection range and real-time detection and distance measurement have great advantages in segmentation, but the classification is low due to the sparse distribution of point clouds. The high resolution camera can classify the target, so the fusion of camera and LIDAR is the most mainstream method when classifying the target. However, road boundaries are different from complex obstacles on the road, it is not necessary to perform accurate classification when extracting road boundaries, and it is only necessary to divide it from the surrounding environment information.

If the road boundary is detected by camera and sensor fusion, the degree of redundancy of the sensor information is increased, which tends to cause the real-time performance of the algorithm to decrease, and the overall detection effect is not good. Due to the differences in the objects being processed and the purposes to be achieved, we have not cited the reference on camera and LIDAR fusion in this paper.

We tried our best to improve the manuscript and made some changes in the manuscript. These changes will not influence the content and framework of the paper. And here we did not list the changes but marked in red in revised paper.

We appreciate for Editors/Reviewers' warm work earnestly, and hope that the correction will meet with approval.

Once again, thank you very much for your comments and suggestions.

Round 2

Reviewer 2 Report

Thank you for properly addressing my comments.

Author Response

We appreciate for your warm work earnestly.

Reviewer 3 Report

ok

Author Response

Thank you very much.